# ZERO-SHOT CROSS-TASK PREFERENCE ALIGNMENT FOR OFFLINE RL VIA OPTIMAL TRANSPORT

## ABSTRACT

In preference-based Reinforcement Learning (PbRL), aligning rewards with human intentions often necessitates a substantial volume of human-provided labels. Furthermore, the expensive preference data from prior tasks often lacks reusability for subsequent tasks, resulting in repetitive labeling for each new task. In this paper, we propose a novel zero-shot cross-task preference-based RL algorithm that leverages labeled preference data from source tasks to infer labels for target tasks, eliminating the requirement for human queries. Our approach utilizes Gromov-Wasserstein distance to align trajectory distributions between source and target tasks. The solved optimal transport matrix serves as a correspondence between trajectories of two tasks, making it possible to identify corresponding trajectory pairs between tasks and transfer the preference labels. However, direct learning from these inferred labels might introduce noisy or inaccurate reward functions. To this end, we introduce Robust Preference Transformer, which considers both reward mean and uncertainty by modeling rewards as Gaussian distributions. Through extensive empirical validation on robotic manipulation tasks from Meta-World and Robomimic, our approach exhibits strong capabilities of transferring preferences between tasks in a zero-shot way and learns reward functions from noisy labels robustly. Notably, our approach significantly surpasses existing methods in limited-data scenarios. The videos of our method are available on the website: https://sites.google.com/view/pot-rpt.

## 1 INTRODUCTION

Recent years have witnessed remarkable achievements in Reinforcement Learning (RL), particularly in addressing sequential decision-making problems given a well-defined reward function (Mnih et al., 2013; Silver et al., 2016; Vinyals et al., 2019; Berner et al., 2019). Nevertheless, the practical application of RL algorithms is often impeded by the considerable effort and time required for reward engineering, along with the unexpected and potentially unsafe outcomes of reward hacking, where RL agents exploit reward functions in unanticipated ways. Furthermore, infusing RL learners with societal norms or human values via crafted reward functions remains a great challenge in certain practical deployment.

As a promising alternative, preference-based RL (Christiano et al., 2017) introduces a paradigm shift from traditional RL by learning reward functions based on human preferences between trajectories rather than manually designed reward functions. By directly capturing human intentions, preference-based RL has demonstrated an ability to teach agents novel behaviors that align more closely with human values. However, while the strides made in preference-based RL are significant (Park et al., 2022; Liang et al., 2022; Liu et al., 2022), current algorithms come with their own set of challenges. First, they are heavily reliant on a vast number of online queries to human experts for preference labels for reward and policy learning. This dependency not only increases the time and cost associated with training but also results in data that cannot be recycled or repurposed for new tasks. Each new task encountered demands its own set of human preference labels, creating a cycle of labeling that is both resource-intensive and inefficient. While Hejna III & Sadigh (2023) leverages prior data to pre-train reward functions via meta-learning and adapts quickly with new task preference data, the need for millions of pre-collected preference labels and further online queries makes this approach impractical in many scenarios.

Recently, Gromov-Wasserstein (GW) distance (Mémoli, 2011) has shown effectiveness in a variety of structured data matching problems, such as graphs (Xu et al., 2019) and point clouds (Peyré et al., 2016). Gromov-Wasserstein distance measures the relational distance and finds the optimal transport

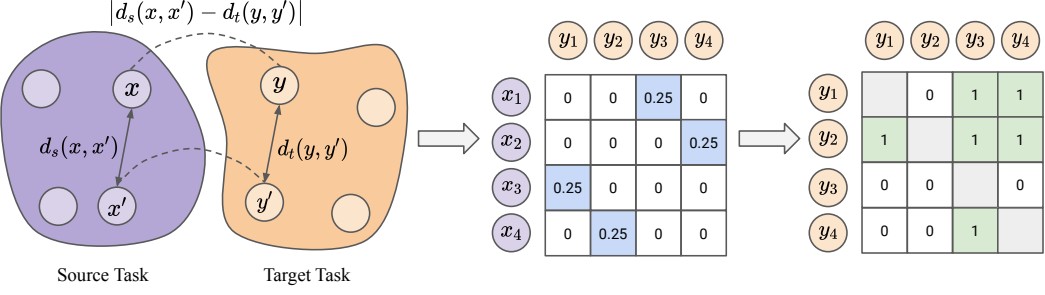

(a) Gromov-Wasserstein Alignment  (b) Solving Transport Matrix  (c) Preference Transfer

Figure 1: Diagram of POT. The circle ◯ represents a trajectory segment in each task. (a) POT uses Gromov-Wasserstein distance as a relational distance metric to align trajectory distributions between source and target tasks. (b) The optimal transport matrix is solved by optimal transport, with each element representing the correspondence between trajectories of two tasks. (c) The preference labels of trajectory pairs of the target task are computed based on trajectory correspondence by Equation 6.

plan across different domains. Inspired by this, we consider using Gromov-Wasserstein distance as an alignment tool between the trajectories of source and target tasks. Given two sets of trajectories from source and target tasks respectively, we can identify the corresponding trajectory pairs between tasks based on the solved optimal transport matrix. Hence, a zero-shot cross-task preference-based RL algorithm can be developed that utilizes previously annotated preference data to transfer the preference labels across tasks.

In this work, we aim to leverage data collected from existing source tasks to reduce the human labeling cost. We propose to use Gromov-Wasserstein distance to find the correspondence between trajectories from source tasks and target tasks and compute preference labels according to trajectory alignment, as shown in Figure 1. Our method only requires a small number of preference labels from source tasks, then obtaining abundant preference labels for the target task. However, the transferred labels may contain a proportion of incorrect labels, which significantly affect reward and policy learning. To learn robustly from POT labels, we introduce a novel distributional reward modeling approach, which not only captures the average reward but also factors in the reward uncertainty.

In summary, our contributions are three-fold. First, we introduce **P**reference **O**ptimal **T**ransport (POT), the first zero-shot cross-task preference-based RL approach that utilizes small amount of preference data from similar tasks to infer pseudo labels via optimal transport. Second, we propose Robust Preference Transformer (RPT) to ensure robust learning from POT labels. Last, we validate the effectiveness of our approach through experiments on several robotic manipulation tasks of Meta-World (Yu et al., 2020) and Robomimic (Mandlekar et al., 2022). The empirical results the strong abilities of our method in zero-shot preference transfer. Moreover, it is shown that our method significantly outperforms current methods when there is a lack of human preference annotations.

## 2 RELATED WORK

**Preference-based Reinforcement Learning.** Preference-based RL algorithms have achieved great success by aligning with human feedback (Christiano et al., 2017; Ibarz et al., 2018; Lee et al., 2021a; Ouyang et al., 2022; Bai et al., 2022). The main challenge of preference-based RL is feedback efficiency and many recent preference-based RL works have contributed to tackle this problem. To improve feedback efficiency, PEBBLE (Lee et al., 2021b) proposes to use unsupervised exploration for policy pre-training. SURF (Park et al., 2022) infers pseudo labels based on reward confidence to take advantage of unlabeled data, while RUNE (Liang et al., 2022) facilitates exploration guided by reward uncertainty. Meta-Reward-Net (Liu et al., 2022) further improves the efficiency by incorporating the performance of the Q-function during reward learning. However, most current preference-based RL methods still requires a large number of human preference labels for training new tasks, and the data cannot be utilized for learning other tasks. To leverage preference data on source tasks and reducing the amount of human feedback, Hejna III & Sadigh (2023) leverages meta learning to pre-train the reward function, achieving fast adaptation on new tasks with few human preferences. Despite the success of Hejna III & Sadigh (2023) in reducing human cost, it still needs 1.5 million labels for pre-training and further online querying for the new task. Recently there is

attention on the offline setting. Preference Transformer (PT) (Kim et al., 2023) proposes to use Transformer architecture to model non-Markovian rewards and outperforms previous methods that model Markovian rewards. IPL (Hejna & Sadigh, 2023) learns policies without reward functions. Nonetheless, PT and IPL still require hundreds of human labels. Our method differs from prior methods that we only need a small number of human labels from source tasks and can obtain extensive preference labels for the new task.

**Optimal Transport.** Optimal Transport (OT) has been widely studied in domain adaptation (Damodaran et al., 2018; Shen et al., 2018), graph matching (Titouan et al., 2019; Xu et al., 2019), recommender systems (Li et al., 2022), and imitation learning (Fickinger et al., 2022). For example, GWL (Xu et al., 2019) is proposed to jointly learn node embeddings and perform graph matching. Li et al. (2022) transfers the knowledge from the source domain to the target domain by using Gromov-Wasserstein distance to align the representation distributions. In the context of RL, there are several imitation learning methods that utilize OT to align the agent's and expert's state-action distributions (Dadashi et al., 2021; Cohen et al., 2021; Haldar et al., 2023a; Luo et al., 2023; Haldar et al., 2023b). For cross-task imitation learning method, GWIL (Fickinger et al., 2022) aligns agent states between source and target tasks and computes pseudo rewards based on solved optimal transport plan. POT is the first preference-based RL algorithm that leverages optimal transport for cross-task learning. POT does not perform representation space alignment, which requires additional gradient computation. It directly uses Gromov-Wasserstein distance to align trajectory distributions between tasks and compute preference labels for the target tasks according to the transport matrix.

**Distributional Modeling for Robust Learning from Noisy Samples.** Traditional representation learning techniques extract features as fixed points. However, such modeling fails to adequately capture data uncertainty, leading to suboptimal performance with noisy data. A series of studies have proposed modeling features as distributions to enhance robustness, seen in person Re-ID (Yu et al., 2019), face recognition (Chang et al., 2020), scene graph generation (Yang et al., 2021), Vision-Language Pre-training (VLP) (Ji et al., 2022). Specifically, these methods utilize Gaussian distributions rather than fixed points to model features, interpreting variance as uncertainty. In preference-based RL, Xue et al. (2023) proposes an encoder-decoder architecture for reward modeling, which encodes state-action features as Gaussian distributions. Consequently, the features can be manipulated in a latent space and they are constrained to be close to a prior distribution to stabilize reward learning process. In our work, we model reward distributions rather than feature distributions and we are the first to model reward distribution in preference-based RL to the best of our knowledge.

## 3 PROBLEM SETTING & PRELIMINARIES

**Problem Setting.** In this paper, we consider preference transfer between tasks share the same action space. We assume there exists a task distribution $p(\mathcal{T})$, with each task $\mathcal{T}$ corresponding to a distinct Markov Decision Process (MDP). MDP is defined by the tuple $(\mathcal{S}, \mathcal{A}, \mathcal{P}, \mathcal{R}, \gamma)$ consisting of a state space $\mathcal{S}$, an action space $\mathcal{A}$, a transition function $\mathcal{P} : \mathcal{S} \times \mathcal{A} \to \mathcal{S}$, a reward function $\mathcal{R} : \mathcal{S} \times \mathcal{A} \to \mathbb{R}$, and a discount factor $\gamma \in (0, 1)$. While the action space $\mathcal{A}$ remain identical across these MDPs, the state space $\mathcal{S}$, the transition function $\mathcal{P}$, the reward function $\mathcal{R}$, and the discount factor $\gamma$ can differ.

In this context, our paper introduces the problem of zero-shot preference transfer. We consider a source task $\mathcal{S} \sim p(\mathcal{T})$ and a target task $\mathcal{T} \sim p(\mathcal{T})$, which means that $\mathcal{S}$ and $\mathcal{T}$ have the same action space. Assume we have $M$ trajectories $x_i$ of task $\mathcal{S}$, $i = 1, \cdots, M$, along with preference labels of all combinations of trajectory pairs $(x_i, x_{i'})$ where $i, i' = 1, \cdots, M, i < i'$. For task $\mathcal{T}$, there are $N$ trajectories $y_j$, $j = 1, \cdots, N$. The goal of our method is to learn a policy $\pi(a \mid s)$ for task $\mathcal{T}$ with preference labels transferred from task $\mathcal{S}$.

**Preference-based Reinforcement Learning.** Preference-based RL is assumed to have no access to the ground-truth reward function and learns a reward function $\widehat{r}_\psi$ from human preferences. A trajectory segment of length $H$ is represented as $x = \{\mathbf{s}_1, \mathbf{a}_1, \cdots, \mathbf{s}_H, \mathbf{a}_H\}$. Given a pair of segments $(x^0, x^1)$, a human provides a preference label $z \in \{0, 1, 0.5\}$, where 0 indicates that $x^0$ is preferred over $x^1$ (denoted as $x^0 \succ x^1$), 1 denotes the reverse preference, and 0.5 indicates the two segments are equally preferable. The preference predictor formulated via the Bradley-Terry model (Bradley & Terry, 1952) is:

$$P_\psi[x^0 \succ x^1] = \frac{\exp \sum_t \widehat{r}_\psi(\mathbf{s}_t^0, \mathbf{a}_t^0)}{\exp \sum_t \widehat{r}_\psi(\mathbf{s}_t^0, \mathbf{a}_t^0) + \exp \sum_t \widehat{r}_\psi(\mathbf{s}_t^1, \mathbf{a}_t^1)}. \tag{1}$$

With a dataset containing trajectory pairs and their labels $\mathcal{D} = \{(x^0, x^1, z)\}$, the parameters of the reward function can be optimized using the following cross-entropy loss:

$$\mathcal{L}_{\text{ce}}(\psi) = - \underset{(x^0, x^1, z) \sim \mathcal{D}}{\mathbb{E}} \left[ (1 - z) \log P_\psi[x^0 \succ x^1] + z \log P_\psi[x^1 \succ x^0] \right].$$ (2)

By aligning the reward function with human preferences, the policy can be learned from labeled transitions by $\hat{r}_\psi$ via RL algorithms.

**Optimal Transport.** Optimal Transport (OT) aims to find the optimal coupling of transporting one distribution into another with minimum cost. Unlike Wasserstein distance, which measures absolute distance, Gromov-Wasserstein distance is a relational distance metric incorporating the metric structures of the underlying spaces (Mémoli, 2011; Peyré et al., 2016). Besides, Gromov-Wasserstein distance measures the distance across different domains, which is beneficial for cross-domain learning. The mathematical definition of Gromov-Wasserstein distance is as follows:

**Definition 1.** *(Gromov-Wasserstein Distance (Peyré et al., 2016)) Let $(\mathcal{X}, d_\mathcal{X}, \mu_\mathcal{X})$ and $(\mathcal{Y}, d_\mathcal{Y}, \mu_\mathcal{Y})$ denote two metric measure spaces, where $d_\mathcal{X}$ and $d_\mathcal{Y}$ represent distance metrics measuring similarity within each task, and $\mu_\mathcal{X}$ and $\mu_\mathcal{Y}$ are Borel probability measures on $\mathcal{X}$ and $\mathcal{Y}$, respectively. For $p \in [1, \infty)$, the p-Gromov-Wasserstein distance is defined as:*

$$\mathcal{GW}(\mu_\mathcal{X}, \mu_\mathcal{Y}) = \left( \inf_{\gamma \in \Pi(\mu_\mathcal{X}, \mu_\mathcal{Y})} \iint_{\mathcal{X} \times \mathcal{Y}, \mathcal{X} \times \mathcal{Y}} L(x, x', y, y')^p d\gamma(x, y) d\gamma(x', y') \right)^{\frac{1}{p}},$$ (3)

*where $L(x, x', y, y') = |d_\mathcal{X}(x, x') - d_\mathcal{Y}(y, y')|$ denotes the relational distance function, and $\Pi(\mu_\mathcal{X}, \mu_\mathcal{Y})$ is the set of joint probability distributions with marginal distributions $\mu_\mathcal{X}$ and $\mu_\mathcal{Y}$.*

## 4 METHOD

In this section, we present Preference Optimal Transport (POT), a zero-shot offline preference-based RL algorithm that transfers preferences between tasks via optimal transport. First, we propose to align the trajectories of source and target tasks using optimal transport and computes preference labels according to the solved optimal alignment matrix. Second, we introduce Robust Preference Transformer (RPT), which additionally incorporates the reward uncertainty by modeling the rewards from a distributional perspective, enabling robust learning from noisy labels.

### 4.1 PREFERENCE OPTIMAL TRANSPORT

Gromov-Wasserstein distance shows great abilities in aligning structural information, such as correspondence of edges between two graphs. Therefore, we consider using Gromov-Wasserstein distance as an alignment metric between the trajectories of source and target tasks, finding the alignment of paired trajectories between tasks, and inferring preference labels based on the correspondence and preference labels of the source trajectory pairs.

POT aims to identify the correspondence between two sets of trajectories and transfer the preferences based on accordingly. In this paper, we consider preference transfer problem from a source task $\mathcal{S}$ to a target task $\mathcal{T}$, with their distributions denoted as $\boldsymbol{\mu}$ and $\boldsymbol{\nu}$, respectively. Assume we have $M$ segments with pairwise preference labels $\{x_i\}_{i=1}^M$ from the source task and $N$ segments $\{y_j\}_{j=1}^N$ from the target task. The trajectories can be represented by the probability measures $\boldsymbol{\mu} = \sum_{i=1}^M u_i \delta_{x_i}$ and $\boldsymbol{\nu} = \sum_{j=1}^N v_j \delta_{y_j}$, where $\delta_x$ denotes the Dirac function centered on $x$. The weight vectors $\{u_i\}_{i=1}^M$ and $\{v_j\}_{j=1}^N$ satisfy $\sum_{i=1}^M u_i = 1$ and $\sum_{j=1}^N v_j = 1$, respectively. The empirical Gromov-Wasserstein distance for aligning trajectories between source and target tasks is expressed as:

$$\mathcal{GW}^2(\boldsymbol{\mu}, \boldsymbol{\nu}) = \min_{\boldsymbol{T} \in \Pi(\boldsymbol{\mu}, \boldsymbol{\nu})} \sum_{i=1}^M \sum_{i'=1}^M \sum_{j=1}^N \sum_{j'=1}^N |d_s(x_i, x_{i'}) - d_t(y_j, y_{j'})|^2 \, T_{ij} T_{i'j'},$$ (4)

where the optimal transport matrix is $\boldsymbol{T} = [T_{ij}]$, $\Pi(\boldsymbol{\mu}, \boldsymbol{\nu})$ denotes the set of all couplings between $\boldsymbol{\mu}$ and $\boldsymbol{\nu}$, $\Pi(\boldsymbol{\mu}, \boldsymbol{\nu}) = \{\boldsymbol{T} \in \mathbb{R}^{M \times N} \mid \boldsymbol{T} \mathbf{1}_N = \boldsymbol{\mu}, \boldsymbol{T}^\top \mathbf{1}_M = \boldsymbol{\nu}\}$, $\mathbf{1}_M$ denotes a $M$-dimensional vector with all elements equal to one, and $d_s, d_t$ represent the distance function in each task, such as Euclidean function or Cosine function.

Upon solving Equation 4, we obtain the optimal transport matrix $\boldsymbol{T}$ representing the correspondence between the trajectories of the two tasks. Each element, $T_{ij}$, indicates the probability that trajectory

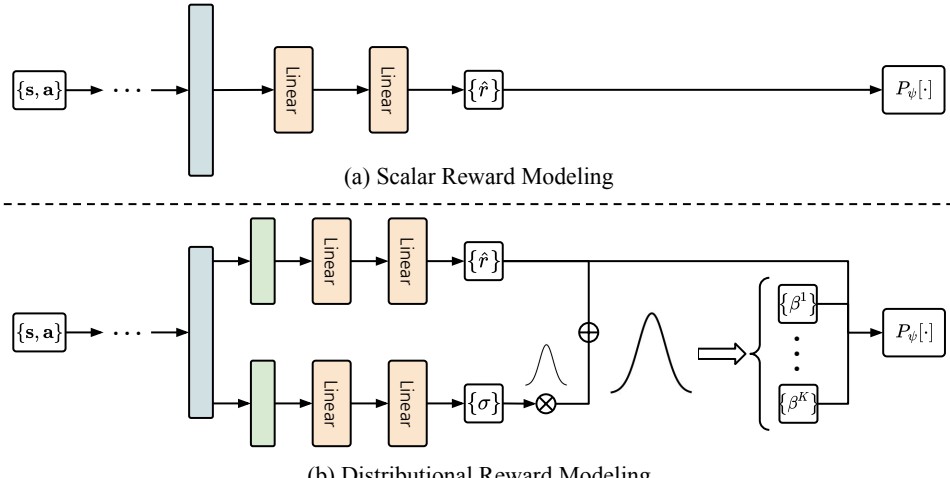

(a) Scalar Reward Modeling

(b) Distributional Reward Modeling

Figure 2: Different types of reward modeling. (a) Scalar reward modeling, which only considers scalar rewards. This modeling type is widely used in preference-based RL algorithms (Christiano et al., 2017; Lee et al., 2021b; Kim et al., 2023). (b) Distributional reward modeling, which adds a branch for modeling reward uncertainty in addition to reward mean.

$x_i$ matches trajectory $y_j$, and the $j$-th column represents the correspondence between $y_j$ and all source trajectories. Therefore, for a pair of trajectories $(y_j, y_j')$, we can identify the paired relations based on the optimal transport matrix. We define the trajectory pair matching matrix $\boldsymbol{A}^{jj'}$ for each $(y_j, y_j')$ by multiplying the $j$-th column $\boldsymbol{T}_{\cdot j}$ and the transpose of $j'$-th column $\boldsymbol{T}_{\cdot j'}^\top$:

$$\boldsymbol{A}^{jj'} = \boldsymbol{T}_{\cdot j}\boldsymbol{T}_{\cdot j'}^\top, \tag{5}$$

where $\boldsymbol{A}^{jj'} \in \mathbb{R}^{N \times N}$, and each element $A_{ii'}^{jj'}$ of the matrix represents the correspondence of trajectory pair $(y_j, y_j')$ with trajectory pair $(x_i, x_i')$ from the source task. If we denote the preference label of $(x_i, x_i')$ as $z(x_i, x_{i'})$, then the POT label of $(y_i, y_{i'})$ is computed as follows:

$$z(y_j, y_{j'}) = \sum_i \sum_{i' \neq i} A_{ii'}^{jj'} z(x_i, x_{i'}), \tag{6}$$

where $i' \neq i$ because the same segments are equally preferable. In Equation 6, the preference labels of source task trajectory pairs are weighted by the trajectory pair correspondence. This means that the preference labels of matched trajectory pairs contribute more to the preference transfer. The full procedures for computing POT labels are shown in Algorithm 1 in Appendix A.

## 4.2 ROBUST PREFERENCE TRANSFORMER

Obtaining preferences labels transferred according to the optimal transport matrix, we can utilize preference-based RL approaches, such as the offline preference-based RL algorithm PT (Kim et al., 2023), to learn reward functions. However, the labels may include some noise and learning from such data using previous methods will influence the accuracy of the rewards and eventually the performance of the policy.

Prior preference-based RL methods represent the rewards as fixed scalar values (Christiano et al., 2017; Lee et al., 2021b; Kim et al., 2023). However, this type of reward modeling is vulnerable to noisy labels. Given a preference dataset comprising trajectory pairs and their preference labels, altering one preference label $z$ of a pair $(x_0, x_1)$ into $1 - z$ will dramatically shift the optimization direction of the reward function on the pair. Thus, if we respectively learn two reward models from the clean dataset and the data with an inverse label, the two reward models will predict distinct values for that trajectory pair. Subsequent, the inaccurate rewards will affect the performance of the policy. Therefore, a robust preference-based RL algorithm capable of learning from noisy labels is necessary.

**Distributional Reward Modeling.** To improve the robustness of preference-based RL in the presence of noisy labels, we incorporate reward uncertainty and model the rewards from a distributional perspective. Specifically, the rewards are modeled as Gaussian distributions, where the mean represents the estimated reward and the variance signifies the reward uncertainty.

As shown in Figure 2, we design two branches for modeling reward mean and variance concurrently. Given the extracted embedding of a trajectory segment represented as $\{\mathbf{x}_t\}$, we split $\{\mathbf{x}_t\}$ into two tensors of the same shape along the embedding dimension. These split tensors are separately processed by the mean and variance branches, ultimately yielding reward mean $\{\hat{r}_t\}$ and variance $\{\sigma_t^2\}$. With reward mean and variance, we then construct the preference predictor $P_\psi$ and derive the loss function for distributional reward learning based on Equation 2:

$$\mathcal{L}_{\text{ce}} = \mathop{\mathbb{E}}_{(x^0, x^1, z) \sim \mathcal{D}} \Big[ \text{CE}\big(P_\psi(\{\hat{r}_t^0\}, \{\hat{r}_t^1\}), z\big) + \lambda \cdot \mathbb{E}_{\beta_t^0 \sim p(\beta_t^0), \beta_t^1 \sim p(\beta_t^1)} \text{CE}\big(P_\psi(\{\beta_t^0\}, \{\beta_t^1\}), z\big) \Big], \quad (7)$$

where $\lambda$ balances the reward mean $\{\hat{r}_t\}$ and the stochastic term $\{\beta_t\}$, $\{\hat{r}_t^0\}$ and $\{\beta_t^0\}$ respectively denote the reward mean and reward samples of trajectory segment $x^0$ (and $\{\hat{r}_t^1\}$ and $\{\beta_t^1\}$ for $x^1$), preference predictor $P_\psi$ in the first term takes the reward mean of two segments as inputs while the second $P_\psi$ uses sampled rewards of two segments as inputs, and CE denotes the cross-entropy loss. In practical, the second expectation in Equation 7 is approximated by the mean of $K$ samples from the distribution of $\beta$.

**Regularization Loss.** The sampled rewards with large variance will make the second term of Equation 7 a large value. If we directly optimize Equation 7, the variance of all samples will decrease, and eventually close to zero. Therefore, to avoid the variance collapse, we introduce a regularization loss to force the uncertainty level to maintain a level $\eta$:

$$\mathcal{L}_{\text{reg}} = \max(0, \eta - h(\mathcal{N}(\hat{r}, \sigma^2))), \quad (8)$$

where $h(\mathcal{N}(\hat{r}, \sigma^2)) = \frac{1}{2} \log(2\pi e \sigma^2)$ computes the entropy of the Gaussian distribution. Combing the cross-entropy loss in Equation 7 and regularization loss in Equation 8, the total loss for RPT training is as follows:

$$\mathcal{L}(\psi) = \mathcal{L}_{\text{ce}} + \alpha \cdot \mathcal{L}_{\text{reg}}, \quad (9)$$

where $\alpha$ is a trade-off factor between the two terms.

**Reparameterization Trick.** Directly sampling $\beta$ from $\mathcal{N}(\hat{r}, \sigma^2)$ will prevent the back propagation process. Hence, we use the reparameterization trick to first sample a noise $\epsilon$ from standard Gaussian distribution $\mathcal{N}(0, 1)$, and computes the sample by:

$$\beta = \hat{r} + \sigma \cdot \epsilon. \quad (10)$$

Therefore, the reward mean and variance can be learned without the influences of sampling operation.

### 4.3 PRACTICAL ALGORITHM

The entire algorithm mainly comprises three stages. First, our approach computes POT labels based on Gromov-Wasserstein distance alignment and the procedures are shown in Algorithm 1 in Appendix A. In Algorithm 1, we use Sinkhorn algorithm (Peyré et al., 2016) to solve the optimal transport matrix, which is implemented by Python Optimal Transport (Flamary et al., 2021). Second, the RPT is trained by POT labels obtained from the first step. Last, we relabel the transitions in the offline dataset using the trained reward function and train offline RL algorithms, such as Implicit Q-Learning (IQL) (Kostrikov et al., 2022). The full procedures are shown in Algorithm 2 in Appendix A.

## 5 EXPERIMENTS

In this section, we first conduct experiments to evaluate our proposed method on several pairs of robotic manipulation tasks from Meta-World (Yu et al., 2020) and Robomimic (Mandlekar et al., 2022) in zero-shot setting. Then we demonstrate our approach significantly surpasses existing methods in limited-data scenarios. Last, we evaluate our algorithm with different choice of cost functions and noise levels.

### 5.1 COMPARED METHODS AND TRAINING DETAILS

The following methods are included for experimental evaluation:

- PT (Kim et al., 2023): The original PT algorithm trained from the preference labels computed by the ground-truth rewards.

- PT+Semi: This baseline combines PT with semi-supervised learning, which is proposed in the online feedback-efficient preference-based RL algorithm SURF (Park et al., 2022). The method infers pseudo preference labels of unlabeled data based on the reward confidence.

- IPL (Hejna & Sadigh, 2023): An offline preference-based RL algorithm that learns policies without modeling reward functions.

- PT+Dis: The baseline is a cross-task preference-based RL algorithm that calculates transferred preference labels simply based on the trajectory similarity between tasks.

- PT+POT (Ours): The method is a zero-shot preference-based RL algorithm that learns PT from preference labels transferred by POT.

- RPT+POT (Ours): The method robustly learns RPT from POT labels by modeling reward distribution.

- IPL+POT (Ours): The method learns from POT labels without reward functions.

**Implementation Details.** All methods are implemented based on the officially released code of PT [1] and IPL [2]. RPT is implemented by replacing the preference attention layer of PT with two branches, each comprising a two-layer Multi-layer Perceptrons (MLP), with the other settings identical to PT. Both PT and PT+Semi utilize scripted labels computed according to ground-truth rewards, which is a common way for the evaluation of preference-based RL algorithms (Lee et al., 2021b; Liu et al., 2022; Kim et al., 2023). PT+POT, RPT+POT and IPL+POT are trained with computed POT labels (zero-shot) or a mixture of POT labels and scripted labels (few-shot). All PT-based methods initially train reward models using the preference data, and the offline RL algorithm IQL (Kostrikov et al., 2022) is used for policy learning following PT. For IPL-based method, policies are directly learned from preferences.

For the Meta-World benchmark, Button Press and Faucet Close serve as source tasks, while Window Open, Door Close, Drawer Open, and Sweep Into are evaluated as target tasks. For Robomimic, we set Square-mh as the source task, and Can-mh and Lift-mh as target tasks. The used tasks and datasets are detailed in Appendix C. We set the segment length as 50 for Meta-World tasks and 100 for Robomimic tasks. For the number of target task preference labels, we provide 100 for Window Open and Door Close, 500 for Drawer Open, Can-mh and Lift-mh, and 1000 for Sweep Into. The Euclidean function is employed as the cost function in the Gromov-Wasserstein distance alignment, with different cost functions discussed in Section 5.4. Regarding RPT learning, the margin $\eta$ in Equation 8 is set to 100 for all experiments, with different margin effects evaluated in Appendix D. The number of samples $K$ in Equation 7 is consistently set to 5. The weight $\lambda$ in Equation 7 is 0.3 for Door Close with Button Press as the source task, and 0.1 for the other task pairs. The trade-off $\alpha$ in Equation 9 is set to 0.02 for Drawer Open with Button Press as the source task, and 0.01 for all other experiments. Detailed network architectures and hyperparameters of all methods and IQL are presented in Appendix C.

The tasks of Meta-World and Robomimic are evaluated through success rate. Each task is conducted with five random seeds, with the mean and standard deviation of success rates reported. Each run evaluates the policy by rolling out 50 episodes at every evaluation step, calculating performance as the mean success rate over these 50 episodes. All experiments are run on NVIDIA GeForce RTX 3080 and NVIDIA Tesla V100 GPUs with 8 CPU cores.

## 5.2 RESULTS OF ZERO-SHOT PREFERENCE LEARNING

Table 1 shows the results on robotic manipulation tasks of Meta-World and Robomimic with different pairs of source and target tasks [3,4]. For the baselines that use scripted preference labels, PT, PT+Semi and IPL yield outstanding performance on the majority of tasks, where PT achieves a mean success rate of 91.7% on Meta-World Tasks and 83.8% across all tasks. The performance of PT+Semi is almost the same with that of PT on Meta-World tasks, but has a drop on Robomimic tasks. For IPL, it outperforms PT and PT+Semi on Robomimic, while its performance on Meta-World is worse than that of them. By transferring preference via OT, POT attains a mean accuracy of 74.9% in computing preference labels across all tasks. PT trained with POT labels realizes a 71.2% success rate, equating to 85.0% of oracle performance (i.e., the performance of PT trained with scripted labels). RPT,

---

[1] https://github.com/csmile-1006/PreferenceTransformer

[2] https://github.com/jhejna/inverse-preference-learning

[3] PT, PT+Semi and IPL do not require preference data from source tasks, so the their results are solely depend on the target tasks.

[4] PT+Sim cannot work by transferring preferences from Square-mh to Lift-mh because the state dimension of these two tasks are different.

Table 1: Success rate of our method against the baselines on robotic manipulations tasks of Meta-World and Robomimic benchmark. The results are reported with mean and standard deviation across five random seeds.

| Source Task | Target Task | PbRL with Scripted Labels | | | PbRL with Transferred Labels | | | | POT Acc. |
|---|---|---|---|---|---|---|---|---|---|
| | | PT | PT+Semi | IPL | PT+Sim | PT+POT | RPT+POT | IPL+POT | |
| Button Press | Window Open | 89.2 ±5.4 | 86.4 ±3.0 | 91.6 ±6.2 | 44.0 ±26.3 | 85.6 ±17.1 | 88.0 ±11.6 | 91.2 ±5.9 | 87.0 |
| Button Press | Door Close | 94.8 ±4.8 | 94.8 ±7.6 | 75.6 ±32.6 | 63.6 ±24.5 | 59.6 ±49.1 | 78.4 ±29.5 | 46.8 ±30.7 | 78.0 |
| Button Press | Drawer Open | 96.6 ±6.1 | 96.8 ±3.3 | 91.2 ±4.1 | 18.0 ±33.0 | 80.8 ±21.0 | 84.0 ±16.0 | 76.8 ±10.4 | 76.6 |
| Button Press | Sweep Into | 86.0 ±8.7 | 88.4 ±5.2 | 73.2 ±6.4 | 48.8 ±34.9 | 77.2 ±11.0 | 80.0 ±6.8 | 76.8 ±7.6 | 69.5 |
| Faucet Close | Window Open | 89.2 ±5.4 | 86.4 ±3.0 | 91.6 ±6.2 | 21.2 ±17.2 | 84.8 ±10.9 | 88.8 ±6.7 | 88.4 ±11.5 | 87.0 |
| Faucet Close | Door Close | 94.8 ±4.8 | 94.8 ±7.6 | 75.6 ±32.6 | 38.8 ±44.8 | 72.8 ±40.9 | 86.4 ±8.2 | 41.6 ±31.5 | 72.0 |
| Faucet Close | Drawer Open | 96.6 ±6.1 | 96.8 ±3.3 | 91.2 ±4.1 | 56.4 ±23.4 | 79.2 ±8.8 | 90.8 ±12.0 | 70.4 ±11.6 | 77.0 |
| Faucet Close | Sweep Into | 86.0 ±8.7 | 88.4 ±5.2 | 73.2 ±6.4 | 14.0 ±20.0 | 71.6 ±17.4 | 75.2 ±6.6 | 81.6 ±7.1 | 68.4 |
| Square-mh | Can-mh | 35.6 ±11.6 | 30.8 ±12.7 | 50.8 ±12.2 | − | 32.8 ±5.9 | 34.8 ±12.1 | 45.6 ±8.2 | 70.0 |
| Square-mh | Lift-mh | 68.8 ±19.2 | 60.8 ±7.3 | 92.4 ±3.3 | − | 62.0 ±19.1 | 74.4 ±23.1 | 81.6 ±6.1 | 63.2 |
| **Average (without Robomimic)** | | 91.7 | 91.6 | 82.9 | 38.1 | 76.5 | 84.0 | 71.7 | 76.9 |
| **Average (all settings)** | | 83.8 | 82.4 | 80.6 | − | 71.2 | 78.1 | 70.1 | 74.9 |

incorporating reward uncertainty in reward modeling, enhances performance to 78.1% across all tasks when trained with POT labels, equivalent to **93.2%** oracle performance. Also, IPL+POT achieves 70.1% success rate across all tasks, which is equal to 87.0% oracle performance. We can conclude that RPT+POT can achieve competitive performance compared with baselines trained with scripted preference labels. Both PT+POT and RPT+POT outperforms PT+Sim by a large margin, and RPT+POT even exceeds PT on the Lift-mh task, demonstrating the powerful capabilities of POT and RPT in zero-shot preference transfer and robust learning.

## 5.3 RESULTS OF FEW-SHOT PREFERENCE LEARNING

The results in Table 1 have shown strong zero-shot transfer ability of POT. To further balance the human labeling cost and algorithm performance, we are interested in how well does RPT+POT perform when there are a small number of preference labels. For fair comparison, we evaluate our method and PT with the same number of scripted preference labels of the target task, across $F_{oracle} \in \{0, 5, 10, 15, 20\}$. Our method additionally obtains $F_{POT} = 100 - F_{oracle}$ POT labels by transferring from the source task. The results in Figure 3 show that RPT+POT significantly outperforms PT when lacking oracle preference labeling, and the advantage becomes more obvious when the number of labels is smaller. Moreover, RPT+POT even exceeds the Oracle PT (i.e., PT with 100 scripted labels) on Window Open task when $F_{oracle} \in \{5, 10, 15, 20\}$. The results demonstrate the excellent performance of our method when oracle labels are hard to obtain, and POT can be used to significantly to reduce extensive human labeling.

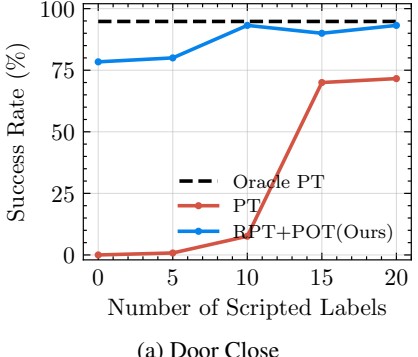

(a) Door Close

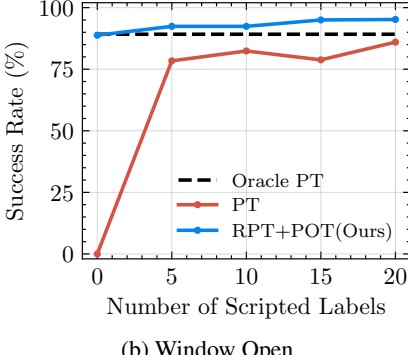

(b) Window Open

Figure 3: Success rate of Door Close and Window Open with different scripted preference labels.

## 5.4 ABLATION STUDY

**Different Cost Functions.** The sensitivity of POT to the cost function is examined by evaluating PT+POT and RPT+POT performance with varying cost functions, including the Euclidean and Cosine functions. Table 2 demonstrates that POT performs robustly with either cost function. Notably, POT

Table 2: Success rates on three pairs of source and target tasks with different cost functions. The results are reported with mean and standard deviation of success rate across five runs.

| Source Task | Target Task | Euclidean | | Cosine | |
|---|---|---|---|---|---|
| | | **RPT+POT** | **POT Acc.** | **RPT+POT** | **POT Acc.** |
| Button Press | Sweep Into | 80.0 ±6.8 | 69.5 | 79.2 ±5.4 | 65.0 |
| Faucet Close | Window Open | 88.8 ±6.7 | 87.0 | 92.4 ±3.6 | 91.0 |
| Square-mh | Lift-mh | 74.4 ±23.1 | 63.2 | 69.3 ±9.5 | 66.0 |
| **Average** | | 81.1 | 73.2 | 80.3 | 74.0 |

with the Cosine function even attains 91.0% accuracy in computing POT labels on the Window Open task, with its success rate (92.4%) surpassing PT with scripted labels on this task (89.2%).

**Different Noise Levels.** To evaluate the performance of PT and RPT under different noise levels, we conduct experiments with 10%, 20%, 30% noisy labels induced by flipping scripted labels. The results in Figure 4 reveal the enhanced robustness of RPT to label noise, with RPT significantly outperforming PT at higher noise levels.

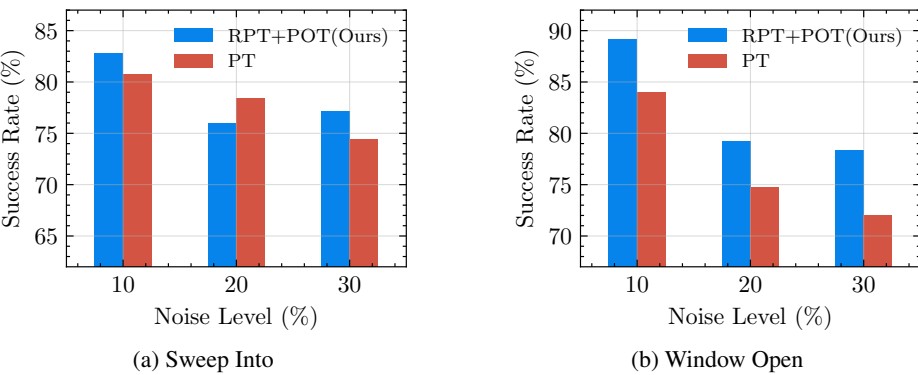

(a) Sweep Into                    (b) Window Open

Figure 4: Success rate of Sweep Into and Window Open under different noise levels.

## 6 CONCLUSION

In this paper, we present POT, a novel cross-task preference-based RL algorithm, which leverages Gromov-Wasserstein distance for aligning trajectory distributions across different tasks and transfers preference labels through optimal transport matrix. POT only needs small amount of preference data from prior tasks, eliminating the need for a substantial amount pre-collected preference data or extensive human queries. Furthermore, we propose Robust Preference Transformer, which models reward uncertainty rather than scalar rewards to robustly learn from POT labels. Empirical results on various robotic manipulation tasks of Meta-World and Robomimic demonstrate the effectiveness of our method in zero-shot transferring accurate preference labels and improves the robustness of learning from noisy labels. Additionally, our method significantly surpasses the current method when there are a few preference labels. By minimizing human labeling costs to a great extent, POT paves the way for the practical applications of preference-based RL algorithms.

**Limitations** Our method does present certain limitations. Firstly, our method is not well-suited for high-dimensional inputs due to the potential for slower processing speeds when working with high-dimensional inputs in optimal transport. Secondly, the efficiency of our algorithm rely on the same action space between source and target tasks. So our method is not suitable for the tasks like those have completely different state and action spaces. A potential solution may be utilizing representation learning methods to obtain trajectory representations and using Gromov-Wasserstein distance to align in the representation space (Chen et al., 2020; Li et al., 2022). We recognize these limitations and view the mitigation of these issues as important directions for future exploration and development.

REPRODUCIBILITY STATEMENT

The source code will be provided in an anonymous repository and we will post it as a comment in the discussion phase. If the paper is accepted, we will open source the code on our website. The experimental details are included in Section 5.1 and Appendix C.

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

## A  ALGORITHM

The algorithm of computing POT labels and the full algorithm of our approach is shown in Algorithm 1 and Algorithm 2, respectively.

---

**Algorithm 1** Computing POT Labels

---

**Input:** source trajectory set $\{x_i\}_{i=1}^M$, target trajectory set $\{y_j\}_{j=1}^N$, regularization parameter $\omega$

1: Initialize $\boldsymbol{T} \leftarrow \frac{1}{MN}\mathbf{1}_M\mathbf{1}_N^\top, \mathbf{p} \leftarrow \frac{1}{M}\mathbf{1}_M, \mathbf{q} \leftarrow \frac{1}{N}\mathbf{1}_N$
2: Compute $\boldsymbol{C}_s, \boldsymbol{C}_t$ with $[\boldsymbol{C}_s]_{ij} = |x_i - x_j|_2$ and $[\boldsymbol{C}_t]_{ij} = |y_i - y_j|_2$
3: Compute $\boldsymbol{C}_{st}$ with $\boldsymbol{C}_{st} \leftarrow \boldsymbol{C}_s^2\mathbf{p}\mathbf{1}_{T_t}^\top + \mathbf{1}_{T_s}\mathbf{q}^\top(\boldsymbol{C}_t^2)^\top$
4: **for** each step **do**
5:     Compute $\boldsymbol{C} \leftarrow \boldsymbol{C}_{st} - 2\boldsymbol{C}_s\boldsymbol{T}(\boldsymbol{C}_t)^\top$
6:     Set $\mathbf{u} \leftarrow \frac{1}{M}\mathbf{1}_M, \mathbf{v} \leftarrow \frac{1}{N}\mathbf{1}_N, \boldsymbol{K} \leftarrow \exp(-\boldsymbol{C}/\omega)$
7:     **for** $k = 1, 2, \cdots$ **do**
8:         $\mathbf{u} \leftarrow \frac{\mathbf{p}}{\boldsymbol{K}\mathbf{v}}, \mathbf{v} \leftarrow \frac{\mathbf{q}}{\boldsymbol{K}^\top\mathbf{u}}$
9:     **end for**
10:    $\boldsymbol{T} \leftarrow \mathrm{diag}(\mathbf{u})\boldsymbol{K}\,\mathrm{diag}(\mathbf{v})$
11: **end for**
12: **for** each $j$ **do**
13:     **for** each $j' \neq j$ **do**
14:         Compute trajectory pair matching matrix $\boldsymbol{A}$ with Equation 5
15:         Compute transferred preference label of $(y_j, y_{j'})$ with Equation 6
16:     **end for**
17: **end for**
**Output:** POT labels

---

---

**Algorithm 2** Robust Preference-based RL from POT Labels

---

**Input:** Source task preference dataset $\mathcal{D}_s$, target task dataset $\mathcal{B}$, reward model learning rate of $\rho$, robust term's weight coefficient $\lambda$, regularization weight coefficient $\alpha$, RPT margin $\eta$, number of reward samples $K$

1: Initialize reward model $\widehat{r}_\psi$, policy $\pi_\phi$, preference dataset of target task $\mathcal{D}_t \leftarrow \emptyset$
2: Perform K-means clustering and group the trajectories of the target dataset into 2 clusters
3: **for** each step **do**
4:     Sample $\frac{N}{2}$ trajectories from each cluster within $\mathcal{B}$
5:     Compute POT labels with Algorithm 1 and store $\mathcal{D}_t \leftarrow \mathcal{D}_t \cup \{(y_j, y_{j'}, z_j)\}_{j=1}^N$
6: **end for**
7: **for** each gradient step **do**
8:     Sample minibatch preference data from $\mathcal{D}_t$
9:     Sample $K$ rewards from the reward distribution computed by the outputs of RPT
10:    Update $\psi$ using Equation 9 with learning rate $\rho$
11: **end for**
12: Label rewards of the transitions in $\mathcal{B}$ using trained $\widehat{r}_\psi$
13: **for** each gradient step **do**
14:     Sample minibatch transitions from $\mathcal{B}$
15:     Update policy $\pi_\phi$ through offline RL algorithms
16: **end for**
**Output:** policy $\pi_\phi$

---

## B  PREFERENCE TRANSFORMER

Preference Transformer (Kim et al., 2023) applies Transformer architecture to model non-Markovian rewards. For a pair of trajectory segments $(x_0, x_1)$, the non-Markovian preference predictor is given by:

$$P_\psi[x^0 \succ x^1] = \frac{\exp\left(\sum_t w_t^0 \cdot \hat{r}_\psi(\{(\mathbf{s}_i^0, \mathbf{a}_i^0)\}_{i=1}^t)\right)}{\exp\left(\sum_t w_t^0 \cdot \hat{r}_\psi(\{(\mathbf{s}_i^0, \mathbf{a}_i^0)\}_{i=1}^t)\right) + \exp\left(\sum_t w_t^1 \cdot \hat{r}_\psi(\{(\mathbf{s}_i^1, \mathbf{a}_i^1)\}_{i=1}^t)\right)}, \quad (11)$$

where $w_t^j = w(\{(\mathbf{s}_i^j, \mathbf{a}_i^j)\}_{i=1}^H)_t, j \in \{0, 1\}$ represents the importance weight. PT introduces a preference attention layer that models the weighted sum of rewards using the self-attention mechanism. Assume the trajectory embedding is $\mathbf{x}_t$, $\mathbf{x}_t$ is projected into a key $\mathbf{k}_t$, query $\mathbf{q}_t$ and value $\hat{r}_t$. The output $z_i$ of self-attention is calculated as:

$$z_i = \sum_{t=1}^H \text{softmax}(\{\langle \mathbf{q}_i, \mathbf{k}_{t'} \rangle\}_{t'=1}^H)_t \cdot \hat{r}_t. \tag{12}$$

The weighted sum of non-Markovian rewards is computed as:

$$\frac{1}{H}\sum_{i=1}^H z_i = \frac{1}{H}\sum_{i=1}^H\sum_{t=1}^H \text{softmax}(\{\langle \mathbf{q}_i, \mathbf{k}_{t'} \rangle\}_{t'=1}^H)_t \cdot \hat{r}_t = \sum_{t=1}^H w_t \hat{r}_t. \tag{13}$$

Obtaining a dataset containing $\mathcal{D} = \{(x^0, x^1, z)\}$, the parameters of PT can be optimized by Equation 2.

## C  EXPERIMENTAL DETAILS

### C.1  TASKS

The used tasks are shown in Figure 5 and the task descriptions are listed as follows:

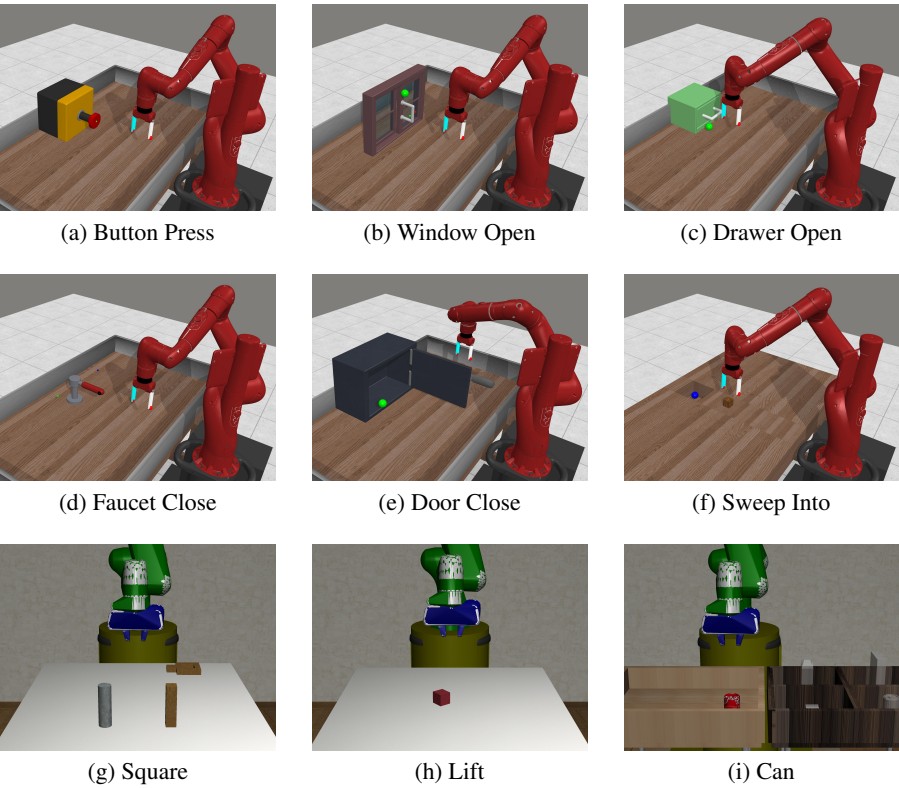

|  |  |  |
|---|---|---|
| (a) Button Press | (b) Window Open | (c) Drawer Open |
| (d) Faucet Close | (e) Door Close | (f) Sweep Into |
| (g) Square | (h) Lift | (i) Can |

Figure 5: Nine robotic manipulation tasks used for experiments. (a-f) Meta-World tasks. (g-i) Robomimic tasks.

**Meta-World.**

- Button Press: The objective is to manipulate a robotic arm to press a button. The button's initial position is arbitrarily placed.

- Faucet Close: The goal is to control a robotic arm to close a faucet. The initial faucet location is randomly assigned.

- Window Open: The task entails commanding a robotic arm to open a window. The window's starting position is randomly chosen.

- Door Close: The task involves guiding a robotic arm to close a door. The door's starting position is selected randomly.

- Drawer Open: The objective is to operate a robotic arm to open a drawer. The drawer's initial placement is arbitrary.

- Sweep Into: The task involves manipulating a robotic arm to propel a ball into a cavity. The ball's initial position is random.

**Robomimic.**

- Square: The goal is to manipulate a robotic arm to lift a square nut and position it on a rod.

- Lift: The task is to operate a robotic arm to elevate a cube to a predefined height.

- Can: The objective is to guide a robotic arm to reposition a can from one container to another.

## C.2 DATASETS

**Meta-World.** For Meta-World tasks, the source preference datasets are collected by ground-truth policies and random policies, with the number of trajectories $M = 4$. For each task, both a ground-truth policy and a random policy are utilized to roll out and obtain 2 trajectories of length 50.

To generate offline dataset for target tasks, we collect the replay buffer and feedback buffer using the preference-based RL algorithm PEBBLE (Lee et al., 2021b). For Window Open and Door Close tasks, PEBBLE is run with 120000 steps and 2000 scripted preference labels. For Drawer Open, we run PEBBLE with 400000 steps and 4000 scripted labels, and for Sweep Into, PEBBLE is run with 400000 steps and 8000 scripted labels.

**Robomimic.** The source dataset of Robomimic tasks is obtained from the Multi-Human (MH) offline dataset of each task, with the number of trajectories $M = 4$. The MH dataset is collected by 6 operators across 3 proficiency levels, with each level comprising 2 operators. Each operator collect 50 demonstrations, resulting in a total of 300 demonstrations. For each task, the source dataset consists of the best 2 trajectories from the offline dataset and 2 random trajectories, and trajectory's length is 100. The offline dataset also serves as the target dataset.

## C.3 IMPLEMENTATION DETAILS

For all task pairs, we first perform K-means clustering and categorize trajectories segments in the feedback buffer into 2 categories, setting $N = 4$. Then we sample 2 trajectories from each category and employ Algorithm 1 to compute POT labels. The detailed hyperparameters of PT and IQL are presented in Table 3 and Table 4, respectively.

Table 3: Hyperparameters of PT.

| Hyperparameter | Value |
| --- | --- |
| Number of layers | 1 |
| Number of attention heads | 4 |
| Embedding dimension | 256 |
| Batch size | 256 |
| Optimizer | AdamW |
| Optimizer betas | $(0.9, 0.99)$ |
| Learning rate | 0.0001 |
| Learning rate decay | Cosine decay |
| Weight decay | 0.0001 |
| Dropout | 0.1 |

## D ADDITIONAL RESULTS

In this section, we conduct additional experiments to evaluate the sensitivity of our method to several critical hyperparameters, which include the robust term's weight coefficient $\lambda$ in $\mathcal{L}_{\text{ce}}$, the

Table 4: Hyperparameters of IQL.

| Hyperparameter | Value |
|---|---|
| Network (Actor, Critic, Value network) | $(256, 256)$ |
| Optimizer (Actor, Critic, Value network) | Adam |
| Learning rate (Actor, Critic, Value network) | 0.0003 |
| Discount | 0.99 |
| Temperature | 3.0 (Meta-World), 0.5 (Robomimic) |
| Expectile | 0.7 |
| Dropout | None (Meta-World), 0.1 (Robomimic) |
| Soft target update rate | 0.005 |

regularization weight coefficient $\alpha$, and the RPT margin $\eta$. The following experiments use Button Press as the source task for Sweep Into, Faucet Close for Window Open, and Square-mh for Lift-mh.

**Robust Term's Weight Coefficient $\lambda$ in $\mathcal{L}_{\text{ce}}$.** The hyperparameter $\lambda$ balances the effects of mean and sampled rewards in Equation 7. To assess the sensitivity of our method to the weight $\lambda$, we perform supplementary experiments with different $\lambda = \{0.001, 0.01, 0.1, 1\}$. The results in Figure 6 demonstrate our method's robustness against $\lambda$ variations.

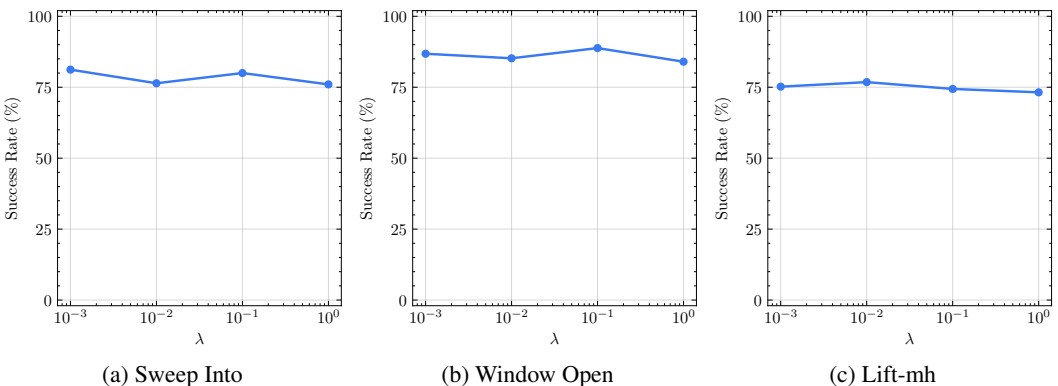

| (a) Sweep Into | (b) Window Open | (c) Lift-mh |
|---|---|---|

Figure 6: Success rate variations of Sweep Into, Window Open and Lift-mh tasks across different $\lambda$ values.

**Regularization Weight Coefficient $\alpha$.** To examine the influence of the weight coefficient $\alpha$ in Equation 9, we evaluate our approach with $\alpha = \{0.001, 0.01, 0.1, 1\}$. As Figure 7 shows, our method retains high success rate with small $\alpha$ values. Conversely, a larger $\alpha$ slightly reduce the performance, as it diminish the contribution of $\mathcal{L}_{\text{ce}}$ to reward learning, which further affects the accuracy of the reward function.

**RPT Margin $\eta$.** $\eta$ serves as a variance constraint in Equation 8. Further experiments are conducted to evaluate this parameter's influence. For Sweep Into, $\eta = \{50, 100, 200\}$ are used for evaluation, while $\eta = \{0.1, 1, 10\}$ are used for Window Open and Lift-mh tasks. The results in Figure 8 demonstrate that our method is not sensitive to the changes of $\eta$.

**Number of Source Task Trajectories $M$.** We additionally conduct experiments to evaluate the effect of the number of source task trajectories to our method, across $M \in \{4, 8, 16\}$. The results in Table 5 show that our method is not sensitive to the number of source trajectories.

**Videos and Demos.** We provide supplementary videos showcasing the trajectories of agents trained using our method for each task pair. Furthermore, we offer several demos illustrating the computation of POT labels on our website. Please visit the website for details.

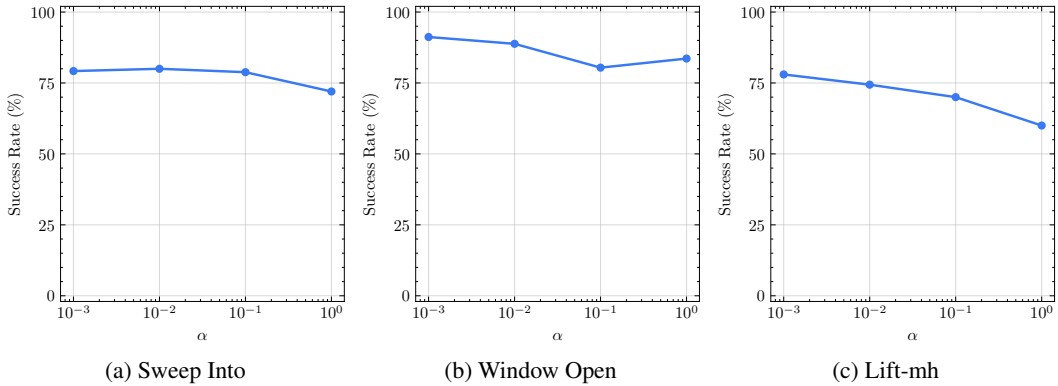

Figure 7: Success rate variations for Sweep Into, Window Open, and Lift-mh tasks across different $\alpha$ values.

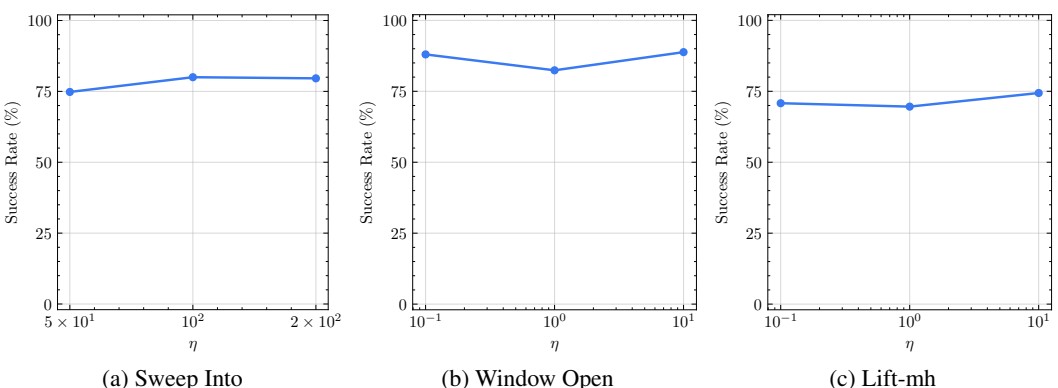

Figure 8: Success rate variations for Sweep Into, Window Open, and Lift-mh tasks across different $\eta$ values.

Table 5: Success rate and accuracy of POT labels on different numbers of source task trajectories. The results are reported with mean and standard deviation of success rate across five runs.

| Source Task | Target Task | $M = 4$ | | $M = 8$ | | $M = 16$ | |
|---|---|---|---|---|---|---|---|
| | | RPT+POT | POT Acc. | RPT+POT | POT Acc. | RPT+POT | POT Acc. |
| Button Press | Drawer Open | $84.0_{\pm 16.0}$ | 76.6 | $82.6_{\pm 17.7}$ | 77.6 | $85.2_{\pm 15.8}$ | 76.6 |
| Faucet Close | Window Open | $88.8_{\pm 6.7}$ | 87.0 | $85.2_{\pm 9.4}$ | 85.0 | $88.4_{\pm 11.0}$ | 87.0 |
| **Average** | | 86.4 | 81.8 | 83.9 | 81.3 | 86.8 | 81.8 |

