# OpenReview forum: "Zero-shot Cross-task Preference Alignment for Offline RL via Optimal Transport"
_ICLR.cc/2024/Conference — Submitted to ICLR 2024_

### Official Review · Reviewer_gYxn · 2023-10-29

**Soundness:** 3 good
**Presentation:** 3 good
**Contribution:** 3 good
**Rating:** 5
**Confidence:** 3

**Summary:**

This paper introduces Preference Optimal Transport (POT), which aims to establish a correspondence between the same human preference in different reinforcement learning tasks. It solves a question: if a human user prefers trajectory x in the source domain, what trajectory y would the same user prefer in the target domain? Specifically, POT aligns two sets of trajectories by solving an optimal transport matrix under Gromov-Wasserstein distance. POT also incorporates uncertainty in human preferences by using distributional rewards.

**Strengths:**

1. Clear identification of the problem (user preference transfer between RL task domains)
2. Clear presentation of the POT algorithm.

**Weaknesses:**

In Table 1, we can see that scripted labels are in general providing better results than transferred labels, which is expected. Therefore, transferred labels should only be used as a substitute when scripted labels are expensive. The paper has yet to discuss applicable scenarios of using transferred labels - when should we compromise success rate for cheaper labels?

**Questions:**

Please respond to the weakness above. If is addressed appropriately, I am willing to improve my rating.

---

### Official Review · Reviewer_43na · 2023-10-31

**Soundness:** 3 good
**Presentation:** 3 good
**Contribution:** 2 fair
**Rating:** 5
**Confidence:** 4

**Summary:**

The paper introduces an algorithm for zero-shot preference-based RL, aiming to address the human preference-guided RL challenge in a transfer learning context. Specifically, the authors utilize optimal transport theory to transfer human preference labels from one RL task to another. This transfer procedure involves computing a coupling matrix using the Gromov-Wasserstein distance, and this matrix yields the transferred preference labels. Additionally, the reward function is modeled as a distribution, facilitating learning from noisy labels and is refined using a process that employs Gaussian distributions instead of scalar rewards. The proposed approach is assessed on robot manipulation tasks in the Meta-World and Robomimic environments. Comparative experiments against baselines and thorough ablation studies further validate the design decisions presented in the paper.

**Strengths:**

- Overall, this paper is well-written, and the proposed idea is presented comprehensively.
- The method proposed uses the Gromov-Wasserstein distance to learn a coupling and subsequently transfers the preference label. This approach allows for the creation of a reward function for a target task using preference labels from source tasks, without the need for preference labels specific to the target task.
- The authors have enhanced the robustness of the RL objective by introducing Gaussian noise.
- Comprehensive experimental results are provided to validate the proposed method.

**Weaknesses:**

- The contribution could be the weakness of this work. It seems that the proposed method is rather incremental since the authors incorporate the original preference-based reinforcement with off-the-shelf Gromov-Wasserstein distance method. In addition, the uncertainty module also utilizes the existing reparameterization trick. Since the Gromov-Wasserstein distance is a concrete way to measure the difference between different distributions, is it possible for the authors can provide generalization bounds (even in the toy example)?
- Some RL tasks can be difficult to generalize. The authors haven't clarified how their optimal transport technique might be effective across such different scenarios.
- The authors may better justify the usage of preference in RL settings. The simulation environments used in this work are not designed to validate human preference, they are more suitable for goal-conditioned RL. The RL task displayed in this work are reply on the end-effector of the robot arm. The authors may want to clarify what is the underlying benefit of transferring human performance.
- It seems that the total number of trajectories is 4, and the authors use a K-means clustering to separate them further. I am wondering about the necessity of doing this. Are you labeling the preference among clusters or within clusters?

**Questions:**

- Task Similarity: It appears that different tasks represent variations of the same RL task, with only differing goals. This seems a restrictive setting. Can the authors discuss the method's versatility?

- Preference Matrix Clarification: Including the original preference matrix would enhance comprehension. Consider the source samples matrix:
​[[/, 1, 0], [0, /, 0], [1, 1, /]], and using this and the coupling matrix: [[1/6, 1/6, 0], [1/6, 1/6, 0], [0, 0, 1/3]]. Could the authors detail the preference transfer?

- Preference Transfer Properties: What are the inherent properties of the proposed method? A deeper analysis using the aforementioned matrices could shed light on this.

**Details Of Ethics Concerns:**

No concerns

---

### Official Review · Reviewer_Lako · 2023-11-01

**Soundness:** 1 poor
**Presentation:** 3 good
**Contribution:** 2 fair
**Rating:** 3
**Confidence:** 3

**Summary:**

In Preference-based Reinforcement Learning, matching rewards with human intentions typically demands a significant amount of labels provided by humans. Moreover, the costly preference data from previous tasks often isn't reusable for future tasks, leading to repeated labeling processes. This paper introduces a zero-shot cross-task preference-based RL method that employs preference labels from labeled data to deduce labels for other data, thus bypassing the need for additional human input. The authors employ the Gromov-Wasserstein distance to map trajectory distributions across tasks. Yet, relying solely on these transferred labels could lead to potentially imprecise reward functions. Addressing this, we present the Robust Preference Transformer. It estimates both the average and variance of rewards by representing them as Gaussian distributions. The proposed methodology, when tested on Meta-World and Robomimic, demonstrates superior performance compared with baselines.

**Strengths:**

The paper is well-written and easy to follow.

**Weaknesses:**

I don't feel the simple addition of labels can work. Let me provide a toy example. If the authors pointed out my error, I would be glad to change my score.

Assume the pair-wise labels of source $\{x_1, x_2, x_3\}$ is $Z_s = \[\[/, 0, 0\], \[1, /, 1\], \[1, 0, /\]\]$.

Let's assume the target is exactly the same as the source, i.e., $\{y_1 = x_1, y_2 = x_2, y_3 = x_3\}$. Then the transportation matrix is $\boldsymbol T = \[\[1/3, 0, 0\], \[0, 1/3, 0\], \[0, 0, 1/3\]\]$

Let's say we want to compute the label of $(y_2, y_3)$

$\boldsymbol A^{23} = \[\[0, 0, 0\], \[0, 0, 1/9\], \[0, 0, 0\]\] $

$z(y_2, y_3)=\frac{1}{9}z(x_2, x_3)=1/9$, while we know the ground truth is $z(y_2, y_3)=z(x_2, x_3)=1$.

In addition, some notations are ambiguous, e.g., $\mathcal S$ for state space and source task at the same time.

**Questions:**

* In Figure 3b, why is RPT+POT even better than Oracle PT? If it is due to the variance, error bars should be provided in all figures.

---

> ### Author Response · Authors · 2023-11-13
> **Response to Reviewer Lako**
>
> We thank reviewer Lako for the constructive comments. We will give our point-wise responses below.
>
> **Q1: A toy example.**
>
> > **A1**: Your result is right and we apologize for the confusion. Actually, we have noticed that most of the computed label values are very close to $0$. And we should pay attention to the relative label values instead of absolute values. Therefore, we adopt a min-max normalization to the matrix $Z_t$, with each element $z(y_j, y_j^\prime)$ representing the POT label of $(y_j, y_j^\prime)$. Also, since the labels are binary in practical implementation, we let the values greater than $0.5$ be label $1$ and make the other values be label $0$.
> > In this toy example, after min-max normalization and binarization, $Z_t = [[/, 0, 0], [1, /, 1], [1, 0, /]]$, which is exactly equal to $Z_s$. We will update the draft to make it clear.
>
> **Q2: "Some notations are ambiguous, e.g., $\mathcal{S}$ for state space and source task at the same time."**
>
> > **A2**: Thanks for pointing out this. We will carefully revise the notations in the updated draft.
>
> **Q3: "In Figure 3b, why is RPT+POT even better than Oracle PT? If it is due to the variance, error bars should be provided in all figures."**
>
> > **A3**: This is an insteresting question. In preference-based RL, the oracle labels are computed based on the ground-truth reward function of each task. One reason may be the reward function is not defined properly. And sometimes the oracle labels cannot reflect the real human intentions, which is pointed out by [1]. So, directly using oracle labels may obtain unsatisfied results. We will update the draft to include the error bars.
>
> **Reference**
>
> [1] Preference transformer: Modeling human preferences using transformers for RL.

---

### Official Review · Reviewer_g2DT · 2023-11-01

**Soundness:** 2 fair
**Presentation:** 3 good
**Contribution:** 2 fair
**Rating:** 3
**Confidence:** 4

**Summary:**

This paper first introduces a zero-shot cross-task transfer algorithm Preference Optimal Transport (POT) for preference-based offline reinforcement learning. The trajectories between the source and target tasks are aligned via the optimal transport method and generates pseudo preference labels based on the alignment matrix. Additionally, the paper introduces the Robust Preference Transformer (RPT) to model the uncertainty of preference labels, enabling robust learning in the presence of transfer noise. Experimental results demonstrate that the proposed algorithm exhibits significant advantages in both zero-shot and few-shot preference.

**Strengths:**

1. The paper addresses a highly meaningful scenario that could have a positive impact in practical applications.

2. The writing logic of the paper is very clear, and it is almost devoid of difficulty in understanding.

**Weaknesses:**

1. There are issues with the problem formulation in the paper. The assumption of identical action space alone is not sufficient to guarantee the alignment of trajectories and preference labels between source and target tasks. The fundamental reasons for the success of Preference Optimal Transport (POT) are not adequately explained.
2. Transferring the preference labels from the source task to the target task undoubtedly involves negative noise or uncertainty. This problem becomes even more severe in the context of zero-shot learning, where there is no corrective information available. While the paper acknowledges modeling uncertainty as variance, it doesn't eliminate this negative impact on the downstream tasks, which is unreasonable. On the contrary, the success of the target task appears to depend on such uncertainty since RPT+POT > PT+POT.

**Questions:**

1. The paper proposes that uncertainty should approach a predefined value $\mu$ during training. What will happen if the uncertainty is directly set to $\mu$ without training?
2. Unclear expression：$\mathcal{S}$ represents state space and source task simultaneously and the expression of $\mathcal{T}$ is also a little confusing in **problem setting**; PT+Dis and PT+Sim in experiments.
3. What is value of $u_i$ and $v_j$ and does the calculation of $A$ need the value of $u_i,v_j$?

---

### Meta-Review · Area_Chair_6ata · 2023-12-05

**Metareview:**

Summary: The paper proposes a zero-shot cross-task transfer algorithm, Preference Optimal Transport (POT), for preference-based offline reinforcement learning. It aligns trajectories between source and target tasks using the optimal transport method and generates pseudo preference labels based on the alignment matrix. The Robust Preference Transformer (RPT) is introduced to model uncertainty in preference labels, enabling robust learning in the presence of transfer noise.

Weaknesses: While the paper addresses a meaningful scenario, reviewers express concerns about the problem formulation, the success of POT, and the impact of uncertainty on downstream tasks. There are also questions about the handling of uncertainty, notation clarity, and the justification for using transferred labels. There are some concerns about relevant scenarios for using transferred labels.

Strengths: The presentation is generally good, but the contribution and soundness receive mixed feedback. The experiments do demonstrate advantages in zero-shot and few-shot preferences.

Suggestions: 1) clarify the fundamental reasons behind Preference Optimal Transport's (POT) success
2) address concerns about its efficacy and the conditions under which it aligns trajectories and preference labels
3) improve notation clarity
4) address questions from reviewers, and provide additional insights into the versatility of the proposed method and its ability to handle and generalize to different RL scenarios.

**Justification For Why Not Higher Score:**

While the paper addresses a meaningful scenario, reviewers express concerns about the problem formulation, the success of POT, and the impact of uncertainty on downstream tasks. There are also questions about the handling of uncertainty, notation clarity, and the justification for using transferred labels. There are some concerns about relevant scenarios for using transferred labels.

**Justification For Why Not Lower Score:**

N/A

---

### Decision · Program_Chairs · 2024-01-16

Reject